# Utilization of Epidural Electrodes as a Diagnostic Tool in Intractable Epilepsy—A Technical Note

**DOI:** 10.3390/mi13030397

**Published:** 2022-02-28

**Authors:** Ran Xu, Johannes Achberger, Dario von Wedel, Peter Vajkoczy, Julia Onken, Ulf C. Schneider

**Affiliations:** 1Department of Neurosurgery, Charité—Universitätsmedizin Berlin, 13437 Berlin, Germany; ran.xu@charite.de (R.X.); johannes.achberger@mailbox.tu-dresden.de (J.A.); dario.von-wedel@charite.de (D.v.W.); peter.vajkoczy@charite.de (P.V.); julia.onken@charite.de (J.O.); 2BIH Charité (Junior) (Digital) Clinician Scientist Program, Berlin Institute of Health at Charité—Universitätsmedizin Berlin, BIH Biomedical Innovation Academy, Charitéplatz 1, 10117 Berlin, Germany; 3Cantonal Hospital of Lucerne, Spitalstraβe 16, 6000 Lucerne, Switzerland

**Keywords:** epidural electrodes, Peg electrodes, epilepsy, invasive diagnostics, Fo electrodes, depth electrodes

## Abstract

The utilization of epidural electrodes in the preoperative evaluation of intractable epilepsy is a valuable but underrepresented tool. In recent years, we have adapted the use of cylindrical epidural 1-contact electrodes (1-CE) instead of Peg electrodes. 1-CEs are more versatile since their explantation is a possible bedside procedure. Here we report our experience with 1-CEs as well as associated technical nuances. This retrospective analysis included 56 patients with intractable epilepsy who underwent epidural electrode placement for presurgical evaluation at the Department of Neurosurgery at the Charité University Hospital from September 2011 to July 2021. The median age at surgery was 36.3 years (range: 18–87), with 30 (53.6%) female and 26 (46.4%) male patients. Overall, 507 electrodes were implanted: 93 Fo electrodes, 33 depth electrodes, and 381 epidural electrodes, with a mean total surgical time of 100.5 ± 38 min and 11.8 ± 5 min per electrode. There was a total number of 24 complications in 21 patients (8 Fo electrode dislocations, 6 CSF leaks, 6 epidural electrode dislocations or malfunction, 3 wound infections, and 2 hemorrhages); 11 of these required revision surgery. The relative electrode complication rates were 3/222 (1.4%) in Peg electrodes and 3/159 (1.9%) in 1-CE. In summary, epidural recording via 1-CE is technically feasible, harbours an acceptable complication rate, and adequately replaces Peg electrodes.

## 1. Introduction

Despite the vast range of pharmacological treatment options, the limited efficacy of antiepileptic drugs remains a challenge in epilepsy patients. Approximately a third of patients suffering from epilepsy cannot be adequately managed with antiepileptic drugs, sometimes necessitating more invasive approaches as a last resort to localize the seizure focus [1,2,3]. Surgical treatment must be preceded by exhaustive and conclusive diagnostics and characterization of epilepsy as well as its underlying cause [4]. A fundamental pillar in the localization of epileptogenic zones comprises intracranial EEG recording.

Several modalities of intracranial EEG recording have been in use in epilepsy surgery, including (Foramen ovale) Fo electrodes, subdural grids, depth electrodes, and epidural electrodes, each presenting with a unique subset of advantages, requirements, limitations, and risks, making an individual evaluation of each patient, and matching the most suitable modality necessary. Being the first modality ever utilized in series for invasive EEG monitoring of epilepsy patients, epidural electrodes have been around since the late 1930s, described by Penfield and Jasper [5]. Less invasive than most other modalities, epidural electrodes are usually arranged as single contact units and can be inserted through burr or twist drill holes [6,7,8,9]. The electrodes are placed in the epidural space, facilitating the monitoring of the underlying cortex. Thus, the implantation process is relatively straightforward, with short operation times and low perioperative risks. However, as shown in Figure 1, epidural electrodes are disproportionally underrepresented in the current literature compared to other intracranial recording procedures.

In the past decade, we have shifted our epidural electrode technique from Peg electrodes to cylindrical 1-contact electrodes (1-CE) due to the expiration of approval of the Peg electrodes. Based on these circumstances, we had to adapt our technique and switch to 1-CEs, which in fact, enabled easier handling, such as their bedside removal without requiring general anesthesia. Here, we present our experience with the utilization of epidural electrodes as a diagnostic tool in intractable epilepsy and elaborate on the technical nuances of this technique.

## 2. Materials and Methods

This retrospective analysis included 56 patients with intractable epilepsy who were operated on at our institution from September 2011 to July 2021. All patients required intracranial monitoring due to the failure of noninvasive methods to localize and, more importantly, lateralize the epileptogenic focus. Operative techniques included the implantation of epidural electrodes, Foramen ovale (Fo) electrodes, depth electrodes, and a combination thereof. In the past years, our practice pattern shifted from implantation of Peg electrodes to epidural 1-CEs (Figure 2).

Between 2015 and 2019, when the use of cylindrical 1-CE emerged, there were also surgical cases in which a combination of these epidural electrodes was utilized. Table 1 shows an overview of the technical characteristics of the 1-CE and Peg electrodes. 

The surgical steps of the implantation of 1-CE are illustrated in Figure 3. Briefly, after general anesthesia, an approximately 1.5 cm skin incision is made, and a small wound retractor is placed (Figure 3A). Afterwards, a diamond drill is used to make a burr hole with a diameter of approximately 5 mm (Figure 3B,C). Then, a small notch is created with a Kerrison punch, into which the electrode can be placed. This step is crucial to assure that the single contact of the electrode is positioned flat on the dura and that the 1-CE stays at its place to avoid peri- or postoperative dislocation of the electrode. The 1-CE is then placed in the notch (Figure 3D,E). A piece of bone wax or gelita sponge is placed over the burr hole to secure the electrode position, and the wound is closed with a non-resorbable 3-0 ethilon stitch. Due to the small diameter size of the electrode, the 1-CE does not have to be tunneled through the skin but instead can be externalized in a transcutaneous fashion (Figure 3F). The outside part of the electrode is then fixated. This technique facilitates the bedside removal of the electrodes without the need for a second anesthesia or reopening of the wound, and hence simplifies the workflow significantly for the patient and treating physicians.

## 3. Results

### 3.1. Patient Characteristics

A total of n = 56 patients underwent epidural electrode placement for presurgical evaluation of intractable epilepsy. The patient characteristics are summarized in Table 2. The median age at surgery was 36.3 years (range: 18–87). A total of 30 (53.6%) of the patients were female and 26 (46.4%) were male. In this patient population, 507 electrodes were implanted with 381 epidural electrodes, of which 222 were Peg electrodes and 159 represented 1-CE electrodes. Patients sometimes also received concomitant placement of Fo electrodes (n_total_ = 93), and depth electrodes (n_total_ = 33). The mean surgical time was 100.5 ± 38 min with a mean time of 11.8 ± 5 min per electrode.

After implantation of epidural electrodes, in 11/56 (19.6%) patients, the diagnosis and lateralization were clear for temporal lobe epilepsy, following temporal lobe resection without further diagnostics. In 16/56 (28.6%) patients, the epidural electrodes posed an intermediate step to guide further invasive diagnostics, including subdural grids (13/56; 23.2%) and depth electrodes, Fo electrodes, or a combination thereof (3/56; 5.4%). For all patients in which subdural grids were placed, the further invasive diagnostics were conclusive, leading to resectioning epileptogenic foci. In 29/56 (51.8%) patients, no further invasive diagnostics or surgical interventions were indicated.

### 3.2. Complications

There were 24 complications in 21 patients in the study population, summarized in Table 3. The complications comprised 8 Fo electrode dislocations (of which two occurred due to iatrogenic removal by the patient), 6 CSF leaks, 6 epidural electrode dislocations or malfunction, 3 wound infections, and 2 hemorrhages on postoperative CT. Of the epidural electrode complications, there were 3 Peg electrode dislocations, 2 1-CE dislocations, and 1-CE malfunction. A total of 7 of the Fo electrode dislocations required revision surgery, while for the epidural electrodes, only 1 Peg dislocation required revision surgery. All 3 wound infections underwent wound revision. The hemorrhages occurred in the depth electrodes only, but were considerably small without neurological symptoms, thus not requiring surgical intervention. 

The relative electrode complication rate categorized for each electrode type was 8.6% in the Fo electrodes, 1.6% in epidural electrodes, and 6.1% in depth electrodes. The complication rate in the old and new epidural electrodes was comparable, with 3/222 (1.4%) Peg electrode dislocations, 2/159 (1.3%) 1-CE dislocations, and 1/159 (0.6%) 1-CE malfunction.

## 4. Discussion

In this study, we report our experience with epidural electrodes in the setting of preoperative monitoring for epilepsy surgery, as well as technical nuances of this technique. In recent years, our technique has shifted from utilizing 1-CE instead of Peg electrodes for epidural electrode placement. This occurred due to an expiring approval of the Peg electrodes that could not be extended by the production company. Due to these circumstances, we adapted the technique with 1-CEs accordingly and found that the new technique enabled easier handling, such as the bedside removal without requiring another general anesthesia or reopening the wound.

Compared to other techniques, such as subdural or depth electrodes, the placement of epidural electrodes respects the integrity of the dura. However, iatrogenic CSF leaks may occur; but in our study cohort, they did not require surgical intervention. Epidural electrodes allow a relatively fast method for placing bilateral electrodes without requiring a craniotomy or repeat surgery for removal. Compared to Peg electrodes, the diameter of the 1-CE is much smaller and thus allows more versatile handing (Figure 4A).

Our data revealed a low complication rate of the epidural electrodes 6/381 (1.6%) in comparison to the complication rate in Fo and depth electrodes (8/93 (8.6%) and 2/33 (6.1%) correspondingly). Specifically, the relative complication rates of the two different epidural electrode types were comparable with 3/222 (1.4%) Peg dislocations, 1/159 (1.3%) 1-CE dislocations, and 1/159 (0.6%) 1-CE dislocation. To avoid dislocation in the 1-CE, we use a small, e.g., 2 mm Kerrison punch to create a “notch” in which the tip of the electrode is placed after the burr hole is made (Figure 4B). After placing the 1-CE, the burr hole is filled with bone wax or gelita sponge so that the electrode stays in its designated space. 

In this series, the relatively high rate of Fo electrode dislocations might be attributed to the following reasons: we do not routinely place stitches to fixate the external part of these electrodes to the skin for cosmetic reasons (to avoid postoperative facial scarring). Hence, a seizure event has a high risk of causing a dislocation in these electrodes. Also, due to the nature of the anatomic location of Fo electrodes, they can sometimes cause intractable trigeminal neuralgic pain or hypesthesia, causing self-removal of the electrodes by the patients, which occurred in 2 of the 8 cases in this series. A combination of wound closure strips (which we routinely use) and medical glue to fixate the Fo electrodes could potentially prevent these dislocations. Our reported rate of hemorrhagic events of the depth electrodes was slightly higher compared to previous studies [10,11,12]. A possible explanation might be due to the circumstance that postoperative CT scans were routinely performed in all cases. These scans were independently and meticulously evaluated in our patient cohort. Even minimal signs of bleeding (in the trajectory of the electrode or on at the entry point), though not clinically relevant, were classified as complications. 

Byrne et al. also reported their experience with epidural cylinder electrodes as an alternative to Peg electrodes, which are similar in shape and configuration to the 1-CE we used herein [13]. In their study, however, the epidural electrodes had multiple contacts and thus, allowed a bigger coverage of the cortical brain surface, presenting more a potential alternative to epidural strip electrodes. As highlighted in their study, the cylindrical epidural electrodes, due to their small associated mass effect, are less likely to cause intracranial hemorrhages than depth electrodes or subdural grids. Especially in the latter, a craniotomy is required and, therefore, carries a higher risk for other postoperative morbidities such as postoperative infections, CSF leaks, hemorrhages, raised intracranial pressure, and bone flap necrosis [11,12,14,15]. However, they do allow a much broader coverage and sophisticated monitoring, particularly when the potential region of interest is located in eloquent areas. 

The armamentarium for intracranial recording in presurgical epilepsy diagnostics comprises several well-established techniques. The decision on invasive diagnostic procedures can be highly individual and largely depends on local preferences and historical evolution. Epidural intracranial recording has been part of this armamentarium ever since diagnostic epilepsy operations have been performed. We, therefore, advocate not to forget about this valuable and minimally invasive tool. In our experience, the placement of 1-CE for epidural recording is a technically feasible alternative to the expiring peg electrodes. They harbor comparable complication rates and even offer some beneficial characteristics. 

## Figures and Tables

**Figure 1 micromachines-13-00397-f001:**
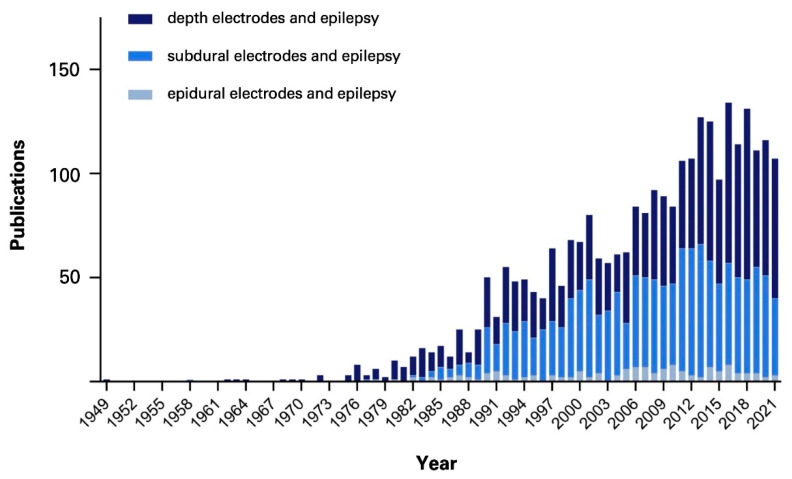
Number of publications per year of the corresponding search queries in PubMed: “depth electrodes and epilepsy”, “subdural electrodes and epilepsy”, and “epidural electrodes and epilepsy”.

**Figure 2 micromachines-13-00397-f002:**
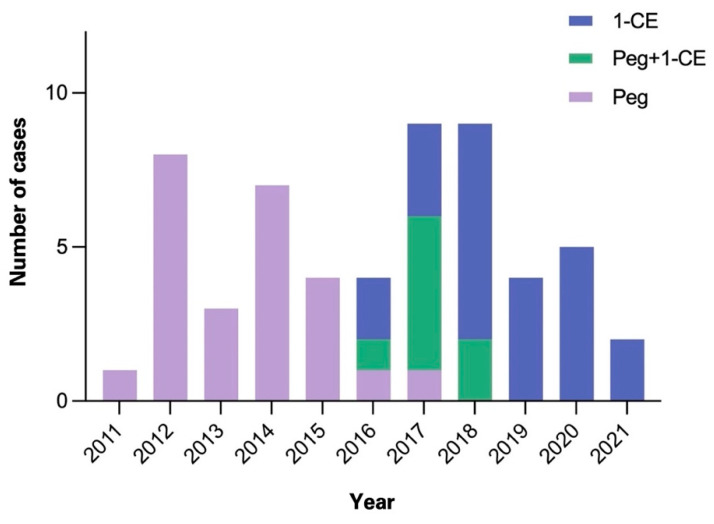
Overview of epidural electrode cases between 2011 and 2021. Number of epidural electrode cases in our institution per annum. Abbreviations: 1-CE = 1-contact electrodes.

**Figure 3 micromachines-13-00397-f003:**
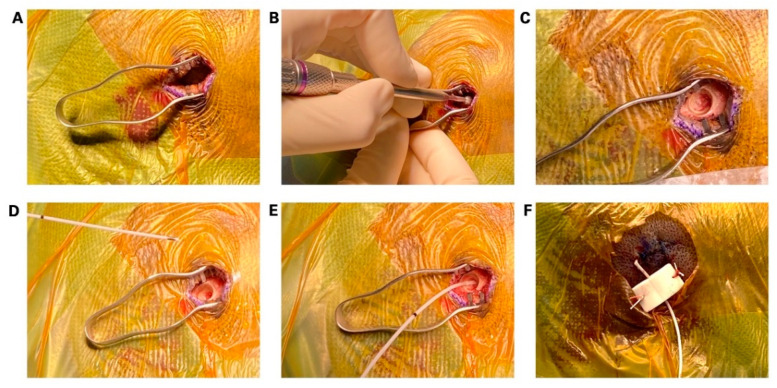
(**A**) step-by-step illustration of the placement of the 1-contact cylindrical electrode (1-CE). A small, approximately 1.5 cm incision is made. (**B**,**C**) A burr hole is made with a small diamond drill. (**D**,**E**) Placement of the 1-CE in a flat position between skull and dura. (**F**) Wound closure with fixation of the epidural electrode.

**Figure 4 micromachines-13-00397-f004:**
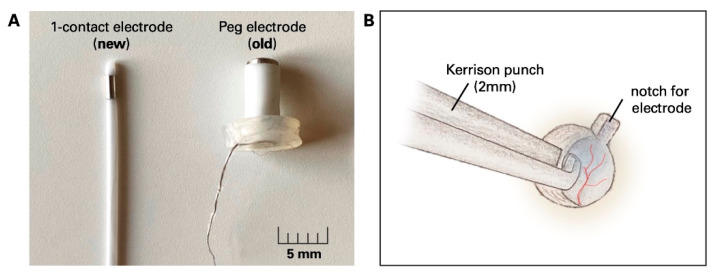
Comparison of epidural 1-CE and Peg electrodes. (**A**) Photograph of the recently implemented 1-CE and old Peg electrode. (**B**) Technical note: a 2 mm Kerrison punch is used to create a small notch in which the epidural electrode is placed to prevent further dislocation in the peri- and postoperative setting. Abbreviations: 1-CE = 1-contact electrode. Scalebar = 5 mm.

**Table 1 micromachines-13-00397-t001:** Technical characteristics of 1-CE and Peg electrodes.

1-Contact Electrodes (1-CE)		Peg Electrodes	
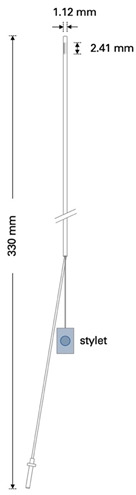	platinum contact cylindrical-shaped removable stylet contact length: 2.41 mm diameter: 1.12 mm total length: 330 mm	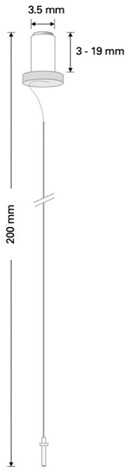	platinum contact mushroom-shaped (material: polyurethane [shaft] and silicone [cap]) length of shaft: 3–19 mm diameter: 3.5 mm total length: 200 mm

**Table 2 micromachines-13-00397-t002:** Patient characteristics (n_total_ = 56).

Age	
median (range)	36.3 (17–57)
**Gender**	
female	30
male	26
diverse	0
**Implanted electrodes**	507
Epidural electrodes	381
Peg (old)	222
3 mm	3
4 mm	34
5 mm	29
6 mm	46
7 mm	30
8 mm	29
9 mm	15
10 mm	24
11 mm	2
12 mm	10
1-CE (new)	159
Fo electrodes	93
Depth electrodes	33
**Surgical time**	
total (mean ± SD)	100.5 ± 38 min
time/electrode (mean ± SD)	11.8 ± 5 min

Abbreviations: Fo = foramen ovale; SD = standard deviation.

**Table 3 micromachines-13-00397-t003:** Overview of complications.

	Complications	Revision Surgery	Electrode
			Complication Rate
**Fo dislocation**	8	7	8/93 (8.6%)
**Dura laceration**	6	0	6/381 (1.6%)
**Epidural electrodes**	6	1	6/381 (1.6%)
**Peg dislocation**	3		3/222 (1.4%)
**1**-**CE**	2		2/159 (1.3%)
**1**-**CE malfunction**	1		1/159 (0.6%)
**wound infection**	3	3	
**hemorrhage**	2		2/33 (6.1%)
from depth electrodes			
**total**	24	11	14/507 (3.2%)

Abbreviations: CSF = cerebral spinal fluid; Fo = Foramen ovale; 1CE = 1-contact electrode.

## Data Availability

All original data of the study will be made available upon request.

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
