# Peer review of "Utilization of Epidural Electrodes as a Diagnostic Tool in Intractable Epilepsy—A Technical Note"

_micromachines, 2022, doi:10.3390/mi13030397_

Round 1

Reviewer 1 Report

The manuscript  “Utilization of epidural electrodes as a diagnostic tool in 2 intractable epilepsy – a technical note” by Dr. Xu is describing the technical aspects of the epidural electrodes, particularly the use of cylindrical epidural 1-contact electrodes (1-CE) instead of Peg electrodes. 

The authors clearly demonstrate the feasibility of 1-CE installation in patients with intractable epilepsy. 

The manuscript reports a thorough study based on the retrospective analysis and is well written. Though the technical advantages of 1-CE are nicely described, the principal point mentioned in the title of the manuscript remains unclear. The manuscript misses justifications that 1-CE electrodes could efficiently replace PEG electrodes in epilepsy tracking. A comparison of the intracranial EEG done by 2 types of electrodes should be shown. This point is crucial, as the results of the comparison will clearly demonstrate the rationale of the 1-CE use. Or manuscript requires reconsidering to show only the technical advantages of 1-CE electrodes installation over PEG electrodes and escaping the question of epilepsy.

Author Response

We would like to thank the reviewer for the consideration of our manuscript. We appreciate the point made here regarding the comparison of intracranial EEG done by the two different types of electrodes. Both 1-CE and PEG electrodes are connected to the same type of EEG device and will show essentially the same EEG. Therefore, it would be redundant to show the EEG of both electrode types in the manuscript. As suggested, though, we shifted the focus on emphasizing only the technical advantages of the 1-CE in the manuscript.

Reviewer 2 Report

Xu et al compared the complications of using Fo electrodes, depth electrodes, Peg electrodes and 1-CE electrodes in a retrospective analysis and made the conclusion that 1-CE electrodes had the acceptable complication rate compared with others. But if they want to emphasize additional advantages of 1-CE electrodes like short operation times and low perioperative risks, more evidence are strongly needed. A table comparing different electrodes characteristics are recommended, which probably is the essential part of this technical note. Also authors may consider to replace “our institution” to the detailed name in the abstract.

Author Response

We would like to thank the reviewer for the careful consideration of our manuscript and thoughtful comments. As suggested, we have added a table in the manuscript in which the technical details of the two electrode types are described and discussed in detail. Furthermore, we have replaced “our institution” to the detailed name in the abstract: “This retrospective analysis included 56 patients with intractable epilepsy who underwent epidural electrode placement for presurgical evaluation at the Department of Neurosurgery at Charité University Hospital from September 2011 to July 2021.“

Round 2

Reviewer 1 Report

In the revised version, the authors incorporated reviewer's comments and stressed the technical advantages of 1-CE electrodes. I have no any comments to the authors of the manuscript and recommend it for publication.